# The association between perceived psychosocial support and resilience among Venezuelan migrant women: A secondary analysis of cross-sectional data from 2022

Maxwell F. Josic[1], Bradley P. Stoner[1], Maria Marisol[2], Susan A. Bartels[1,3]*

**1** Department of Public Health Sciences, Queen's University, Kingston, Canada, **2** International Organization for Migration, Pacaraima Office, Pacaraima, Brazil, **3** Department of Emergency Medicine, Queen's University, Kingston, Canada

* susan.bartels@queensu.ca

## Abstract

Migrants experience profound threats to their mental health, with women facing additional vulnerabilities such as sexual exploitation and trafficking. Resilience protects against the impacts of these threats through mental, emotional, and behavioural adaptations. A central component of resilience is perceived psychosocial support (PPS), which describes the belief that assistance is available to mitigate the effects of stressors. This study analyzes the association between PPS and resilience among Venezuelan migrant women using data from a cross-sectional study (2022) involving 9116 Venezuelan migrants aged 14 + . We hypothesized PPS and resilience would be positively correlated. Following the 'sensemaking' methodology, each participant shared a brief experience and completed a questionnaire contextualizing their experience. PPS and resilience were assessed using two single-item measures: one capturing how supported participants felt, and the other evaluating how often they believed they successfully coped with challenges. Using data from 5388 micro-narratives, we constructed a logistic regression model using backward elimination with inclusion at p < 0.20. Overall, 65% of participants self-reported resilience. The model included five of eight covariates: age, ethnicity, health issues, displacement duration, and relative wealth. Participants in the top tertile of PPS had 2.12 times the odds of resilience compared to those in the bottom tertile (95% CI: [1.84, 2.47], p < 0.0001), while the middle and bottom tertiles were equally resilient (OR=0.99, 95% CI: [0.87, 1.14], p = 0.91). Resilience correlated positively with age and relative wealth, and negatively with displacement duration and health issues. This study confirms PPS is important in the resilience of Venezuelan migrant women and elucidates several unexpected results for further investigation, including the null association between resilience and LGBTQ+ self-identification. Future studies should administer validated questionnaires to better understand the contributions of the constituent components

**Data availability statement:** Data is available in a public, open access repository from: https://doi.org/10.5683/SP3/WPBQB3. The SAS code used in this investigation is available as a Supporting Information file (S5 File. SAS Code).

**Funding:** The original study (2022) was supported by Elrha's Humanitarian Innovation fund (https://www.elrha.org/programme/hif), grant number 48096, received by SAB. Funding for the current investigation was awarded to SAB from the Canada Research Chairs Program (https://www.chairs-chaires.gc.ca/home-accue-il-eng.aspx). The funders had no role in study design, data collection and analysis, decision to publish, or preparation of the manuscript.

**Competing interests:** The authors have declared that no competing interests exist.

of resilience among this population. These results can be utilized to develop tailored resilience-fostering interventions and more efficiently direct mental health resources.

## Introduction

The Venezuelan migration crisis is one of the most significant humanitarian challenges in recent years, with an estimated 7.3 million Venezuelans having fled the country [1] due to severe economic collapse, political instability, and a dramatic decline in public services [2]. Since 2014, Venezuelan migrants have been forced to seek refuge primarily in neighboring Latin American countries, such as Colombia, Peru, Ecuador, and Brazil. These host nations now face unprecedented challenges in providing social, economic, and health services to large volumes of incoming migrants.

Migration is known to heighten vulnerability to mental health threats. Venezuelan migrants are, unfortunately, no exception, and recent research confirms a heightened prevalence of mental health issues among displaced individuals, such as depression and post-traumatic stress disorder [3]. Migrants may experience threats to their mental well-being from a combination of stressors occurring before, during, and after migration. Pre-migration factors include exposure to economic deprivation, food insecurity, trauma, and the emotional toll of leaving behind family and friends. Migration itself often involves trauma from dangerous travel routes and the inherent uncertainty of resettlement. Post-migration challenges, such as xenophobia, social isolation, economic marginalization, and limited access to social and health services, further exacerbate these mental health risks. Long-term exposure to such stressors is associated with a range of negative psychological outcomes [4,5], including fear, anxiety, hopelessness [6,7], post-traumatic stress, depression, anxiety, and problem externalization [8–12]. Women are particularly vulnerable to gendered threats, such as intimate partner violence [13,14], sexual assault, early unions, sexual exploitation, survival and transactional sex, and human trafficking [15]. For instance, the border between Ecuador and Colombia is a frequent corridor for criminal groups involved in the sexual exploitation of women and girls [16], and many Venezuelan migrant women residing in Peru face hyper-sexualization and are stigmatized through accusations of prostitution [17]. The potential harm posed by these gendered threats underscores the importance of critical protective factors, such as resilience, for the mental well-being of Venezuelan migrant women.

Resilience, commonly defined as the ability to adapt and maintain psychological well-being in the face of significant stress, can serve as a powerful buffer against mental health deterioration among migrants. Despite significant heterogeneity regarding which specific factors constitute resilience, the literature overall suggests that, rather than being mostly comprised of relatively stable individual traits [18–22], resilience is a predominantly dynamic process [23–27] in which personal and social resources are mobilized during interactions between individuals and their environments [28] in order to regulate emotions, foster positive coping strategies, and develop adaptive responses. Migrants often face intersecting challenges that all contribute to ongoing insecurity and instability, such as social exclusion, economic instability, and restrictive immigration

policies [29]. For Venezuelan migrant women, resilience may be shaped by complex intersections of cultural background, social support networks, and the uniqueness of each nations' response to the crisis. However, it is important to note that, despite the significantly elevated rate at which they experience adversity and stress, the majority of migrants navigate the challenges of migration and resettlement well and do not develop mental illnesses. Despite this, poor mental health outcomes nonetheless remain disproportionately prevalent among migrants compared to the general population [3], therefore understanding how various factors contribute to the resilience of Venezuelan migrant women is essential for addressing their psychological needs and developing supports for their mental well-being.

Social support is widely recognized as a key contributing factor to resilience [28,30–41] particularly within populations facing displacement and resettlement [42–49], as the migration process often places significant strain on individuals' social networks and previous coping strategies. Perceived psychosocial support (PPS) specifically refers to the emotional, practical, and social resources that individuals *believe* are available to them through their support networks [50–52], which can include family, friends, coworkers, community organizations, and societal institutions, among others. PPS encompasses various types of support, such as practical support (assistance with daily tasks), tangible support (provision of resources like food or money), and emotional support (reassurance, empathy, and advice) [53]. Interestingly, research shows that both received support and *perceptions* of available support each correlate positively with resilience [28,30–40,42–45,47–49,54], indicating that the mere subjective feeling of support being available can be crucial in how individuals cope with adversity. This link between social support and resilience has been confirmed across numerous contexts, as it constitutes myriad forms of assistance for individuals experiencing stress. Studies on migrant populations suggest that strong social support networks bolster resilience by—for example—improving optimism and solidarity among family members [55], developing an individual's self-confidence [56] and coping ability [51], and providing validation, reassurance, empowerment, and a sense of security and belonging [57]. This bolstered resilience, in turn, is associated with improvements in psychological well-being [58], such as increases in hope [46,55], self-confidence [56], emotional intimacy [56,59,60], and the use of healthy coping strategies [55]. By promoting resilience, which acts as a protective buffer, social support networks can empower individuals to successfully overcome mental health threats associated with migration, such as isolation and stress, while also facilitating access to practical resources and fostering emotional stability.

Venezuelan migrant women face heightened exposure to threats to their well-being, such as hyper-sexualization [17] and police extortion [61]. This, in combination with the complex and prolonged nature of the migration crisis, necessitates efforts to explore how PPS contributes to their resilience. Furthermore, investigators should strive to characterize any unique facilitators or barriers these women may face in accessing support, such as legal restrictions, economic hardship, and cultural dissonance, in order to minimize the impacts on their resilience and mental health trajectories.

Despite the critical role PPS plays in promoting resilience, there is a lack of research examining these dynamics within the particular context of the Venezuelan crisis. Rather, most extant resilience studies focus on displaced individuals from vastly different regional and cultural contexts, such as those from Somalia and Syria, thus likely limiting the applicability of their findings to Venezuelan migrant women. For instance, comparatively little is known about the ways in which religion factors into the migration and resettlement process across Latin America. Additionally, while studies on crises revolving around armed conflicts are plentiful, geopolitical and economic crises, such as the one occurring in Venezuela, remain understudied. This issue is further compounded by other broad gaps in the literature, including an alarming deficit of large studies that focus exclusively on women or on South American populations in general. Addressing these gaps is essential for the development of targeted interventions that can bolster the resilience, and thus mental well-being, of Venezuelan migrant women and aid in both their successful integration into host communities and their likely return to Venezuela.

## Objectives

This study aims to address these deficiencies by examining the relationship between PPS and resilience among Venezuelan migrant women in Brazil, Ecuador, and Peru. Through this investigation, we hope to inform targeted public health

policies and interventions that strengthen support networks and resilience among Venezuelan migrant women as they navigate the challenges of displacement and resettlement.

## Methods

### Ethics statement

The original dataset contained no unique identifiers, thus the data used in this study was fully de-identified such that no individual participant could be linked to their responses. Participation was voluntary and informed, with consent having been obtained from each participant by asking them to tick a consent box on the tablet. Given that many adolescent refugees/migrants were without guardians, and some were traveling with their own partners and children, potential participants aged 14–17 and traveling alone were considered to be mature minors, and thus parental approval was deemed unnecessary for their participation. This enabled their full participation in the original study while minimizing the potential for parental involvement to bias the results [62]. Finally, no compensation was provided for participation, and the time commitment required was minimal (~15 minutes). Ethics clearance from the Queen's University Health Science Research Ethics Board was acquired before data was accessed and the investigation was initiated (protocol #6033064).

### Study design

This is a secondary analysis of data collected between January and April 2022 via a quantitative/qualitative, cross-sectional study codesigned by the International Organization for Migration (IOM) and Queen's University [29]. This source investigation examined the gendered migration experiences of Venezuelan migrant women in Brazil, Ecuador, and Peru.

### Sampling

The source study involved trained research assistants interviewing 9116 Venezuelan migrants, aged 14 and above, at nine locations across Brazil (Boa Vista, Manaus, Pacaraima), Ecuador (Huaquillas, Manta, Tulcan), and Peru (Lima, Tacna, Tumbes). Convenience sampling was utilized at all sites, though some snowball sampling also occurred in Lima, where migrants were already well-integrated into the host community—and thus were more difficult to identify. Recruitment was conducted in public spaces frequented by migrants, such as border crossings, points of aid distribution, and refugee/migrant shelters. Additionally, members of equity-deserving groups, such as those identifying as having disabilities and/or being Indigenous, racialized, and/or LGBTQ+, were intentionally recruited through community-based organizations which support them. For the present study, we excluded participants who (a) did not complete both the PPS and resilience questions, (b) shared experiences that were not autobiographical or (c) did not identify as women (cisgender or transgender), consistent with the study's focus on women's resilience.

### Sensemaking methodology

Sensemaking [63] is a mixed quantitative/qualitative narrative-capture methodology with previous use within the Venezuelan migration crisis [29,64,65]. It emphasizes participants' knowledge by empowering them to share, reflect on, and contextualize their own experiences [66–68]. In the original 2022 study, participants were each asked to share a brief 'micro-narrative' on the migration experiences of Venezuelan women, before completing a questionnaire which prompted them to reflect on their shared experience and provide additional contextual and demographic information. As it obtains both quantitative and qualitative data, this methodology enables a more nuanced analysis of complex issues than either data type would in isolation. Since qualitative interviews are detailed, yet resource intensive, while quantitative surveys are comparatively simple, albeit shallow, the integration of survey questions into brief interviews balances depth and efficiency, thereby facilitating increased sample sizes with minimal loss of detail. Sensemaking methodology is also robust

against interpretation and social desirability biases, as participants each interpret their own experiences, and response options are either all positive, all negative, or all neutral, thereby obfuscating a clearly most socially desirable response.

## Survey

The original survey (S2 Appendix) contained two distinct sections. The first required five-to-ten minutes and involved participants responding to their choice of three open-ended prompts related to the migration experiences of Venezuelan women—either by recording audio or by typing about an experience centered around either themselves or someone they knew. The second section also required five-to-ten minutes to complete, and consisted of 31 questions, in several formats, relating to their micro-narratives. Five 'triad' questions had participants plot their responses along a triangular plane, with a response option at each vertex. Four 'dyad' questions permitted a value along a continuous line between two extremes. One 'star' question prompted participants to rate their perspectives on seven metrics, using a two-dimensional grid with 'financial security' along the x-axis and 'well-being' along the y-axis. Because participants could provide micro-narratives in either the first or third person, contextual data was collected using one free-text prompt and twenty multiple-choice items addressing both the respondent and the individual(s) central to the micro-narrative. To ensure full data on the respondents was available for all questions, the current analysis included only participants who shared autobiographical micro-narratives.

The survey questions were initially conceived in English before being translated into Spanish by a professional translator, with accuracy confirmed via back-translation and through pilot testing with 25 Venezuelan women. Feedback from the pilot was used to refine language and improve question clarity prior to the study proper.

## Variables

PPS was quantified through the dyad question "The woman/girl in the shared story was…" which allowed values along a continuous range from "provided with absolutely no supports/services" to "provided with too many supports/services". Responses were collapsed into tertiles ("Bottom", "Middle", and "Top") since a preliminary analysis indicated a unique distribution with potentially interesting patterns at the center and extremes that were important to capture. Social support correlates with resilience across a variety of populations and contexts, including among those displaced by humanitarian crises worldwide [42,44,45,69–71]. Resilience was likewise quantified through the single-response item, "At this time, I am able to cope with the challenges I face", which permitted four choices of response: 'All of the Time', 'Most of the Time', 'Some of the Time', and 'Never'. For this analysis, responses were dichotomized as "All of the Time" versus "Not All of the Time" to yield a clear binary indicator for subgroup comparisons and to guide intervention planning across the full spectrum of resilience. Coping, the cornerstone of resilience [28,32,72], involves action regulation under stress, including the coordination, mobilization, energization, direction, and guidance of behaviour and emotion in response to stressors [73–76], in order to regulate emotions, distress, and the distressing problem itself [77] through purposeful action [78]. Since the original dataset lacked the relevant detail to comprehensively evaluate resilience, this investigation utilized coping as a proxy, as is common throughout the literature [28]. Details on the eight potential confounders (age, ethnicity, having a child, having a partner, identifying as LGBTQ+, length of displacement, miscellaneous health issues, and relative wealth) can be found in S1 Appendix.

## Statistical analysis

Using SAS statistical software (SAS® 9.4 TS1M3), the first phase of the analysis produced descriptive statistics, including response frequencies and measures of spread, for the exposure, outcome, and eight potentially confounding variables. Following this, the correlations between resilience and each individual variable were elucidated in the bivariate analysis phase, which involved one point-biserial and eight chi-squared tests. Then, simple logistic regression modelling yielded unadjusted odds ratios (ORs) and their corresponding 95% confidence intervals between resilience and each covariate.

Finally, a multivariate logistic regression model was constructed to elucidate each variable's contribution to the PPS-resilience relationship. Variables were removed using backward elimination, with a generous inclusion threshold of $p < 0.20$ to account for the high variability characteristic of resilience research, as well as the lack of prior data on both the resilience of South American migrant women and the use of p-value-based selection processes in studies on the resilience of migrants. From this final, parsimonious model, an effect estimate, standard error, OR, 95% confidence interval, and p-value were computed for each response category with respect to their reference values. As guided by previous literature, no interaction terms were included in the model.

Abstinent responses were converted into missing values and, to minimize additional artificial alterations to the original dataset, missing data was used without modification for the descriptive and bivariate analyses. Conversely, to streamline interpretations of the OR estimates, complete case analysis was employed for the logistic regression models, as removing incomplete response sets had a negligible effect on the data.

## Results

The dataset initially contained 9339 narratives shared by 9116 unique participants, as participants could describe more than one experience. As illustrated in Fig 1, 2480 non-autobiographical narratives were removed, as were 634 narratives shared by participants who did not identify as women. Furthermore, 235 response sets which did not contain answers to both the PPS and resilience questions were removed, yielding a sample of 5990 data points for the descriptive and bivariate analyses. The logistic regression model included 5388 response sets, as 602 were incomplete.

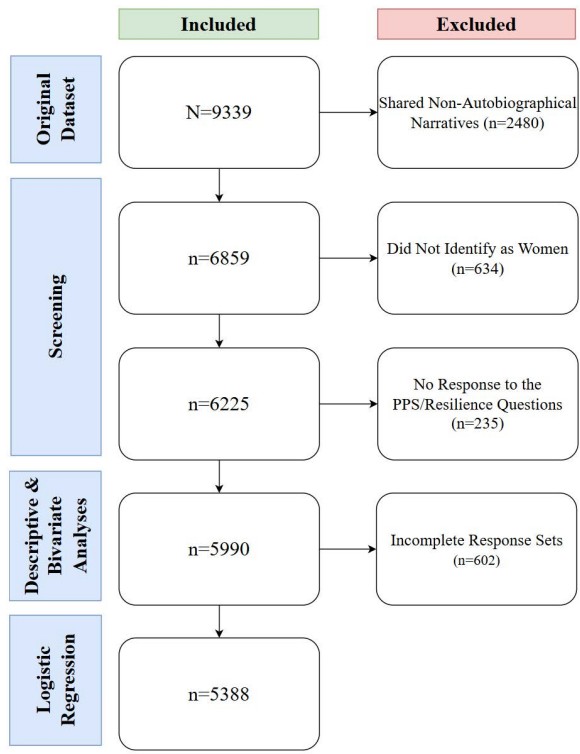

**Fig 1. Subsample selection from parent dataset.**

## Descriptive statistics

Descriptive statistics for the 5990 response sets analyzed are summarized in Table 1. 65% of participants reported being able to cope 'All of the Time', while <1% were 'Never' able to cope. For the PPS dyad question, responses were clustered around 0.2 and 0.75, and were uncommon around 0.5, while the most frequent values were 0 and 1. After excluding the 0.32% of responses with missing data and the three outlying values well-below the minimum participation age of 14 years,

**Table 1. Survey response frequencies.**

| Variable | Response | Total (n = 5990) | Frequency (%) |
|---|---|---|---|
| **Resilience** | All of the Time | 3872 | 64.6 |
| | Most of the Time | 1227 | 20.5 |
| | Some of the Time | 842 | 14.1 |
| | Never | 49 | 0.8 |
| **Ethnicity** | Mestiza | 3309 | 55.2 |
| | Afro-Descendant | 342 | 5.7 |
| | Indigenous | 272 | 4.5 |
| | Other | 56 | 0.9 |
| | None | 1657 | 27.7 |
| | Prefer Not to Say/Not Sure/Missing | 354 | 5.9 |
| **Having a Child** | 0 | 1002 | 16.7 |
| | 1 - 2 | 2742 | 45.8 |
| | 3+ | 2206 | 36.8 |
| | Prefer Not to Say/Missing | 40 | 0.7 |
| **Having a Partner** | Single | 2551 | 42.6 |
| | Married/In a Union | 2818 | 47.1 |
| | Divorced/Separated | 400 | 6.7 |
| | Widowed | 175 | 2.9 |
| | Prefer Not to Say/Missing | 46 | 0.8 |
| **Identifying as LGBTQ+** | Yes | 161 | 2.7 |
| | No | 5676 | 94.8 |
| | Prefer Not to Say/Not Sure/Missing | 153 | 2.6 |
| **Length of Displacement** | < 1 Year | 2869 | 47.9 |
| | 1 - 3 Years | 1783 | 29.8 |
| | 3 - 5 Years | 1129 | 18.9 |
| | > 5 Years | 148 | 2.5 |
| | Prefer Not to Say/Not Sure/Missing | 61 | 1.0 |
| **Miscellaneous Health Issues** | Alcohol and Drugs | 34 | 0.6 |
| | Disability | 412 | 6.9 |
| | Mental Health Issues | 96 | 1.6 |
| | None | 5448 | 91.0 |
| **Relative Wealth** | Very Wealthy | 2 | 0.0 |
| | Wealthy | 19 | 0.3 |
| | Average | 1986 | 33.2 |
| | Poor | 2981 | 49.8 |
| | Very Poor | 779 | 13.0 |
| | Prefer Not to Say/Not Sure/Missing | 223 | 3.7 |

age was normally distributed with a right skew, a mean of 32.7 years, and a median of 30 years. The interquartile range was 15 years and the oldest participant was 89 years old. Longer periods of displacement were decreasingly frequent, falling from 48% of women being displaced for <1 year to only 2.5% for >5 years. Only 9% of participants reported using alcohol and drugs, having a disability (as defined by the individual), or experiencing mental health issues, and 2.7% self-identified as LGBTQ+. The number of children reported varied, while most women either had a partner (47%) or were single having never been married (43%). Finally, nearly all participants considered themselves financially average (33%), poor (50%), or very poor (13%) in comparison to others in their communities. As such, to address the small cell sizes in the subsequent analyses, the remaining 0.35% of participants were combined with those who considered themselves to possess average wealth.

Table 2 presents comparisons between the dichotomized resilience groups. 65% of participants reported being able to cope all the time (the 'high resilience' group), while 35% reported otherwise (the 'low resilience' group). After stratifying the data by resilience, it was far more common for high resilience women to be in the top PPS tertile compared to those with low resilience (39% vs. 23%, p<0.0001). High resilience women were also more likely to be older (r=0.0715, p<0.0001) and displaced for <1 year (51% vs. 44%), while low resilience was more likely for all longer periods of displacement (p<0.0001). Additionally, high resilience was more predominant among those with children (84.1% vs. 81.4%, p=0.009) and partners (48.4% vs 45.6%, p=0.042). Despite small cell sizes, using alcohol and drugs, having a disability, or experiencing mental health issues were consistently more common for women in the low resilience group (p<0.0001). Finally, differences between the two groups were minimal regarding participants' ethnicities (p=0.19) and whether they self-identified as LGBTQ+ (p=0.96).

## Logistic regression modelling

The unadjusted logistic regression performed between PPS and resilience yielded an OR of 1.01 (p=0.89) between the middle and bottom PPS tertiles, and of 2.17 (p<0.0001) between the top and bottom. In the full model, three variables that exceeded the p<0.20 inclusion threshold were removed in the following order: identifying as LGBTQ+ (p=0.47), having a child (p=0.34), and having a partner (p=0.22). All remaining covariates (age, ethnicity, length of displacement, miscellaneous health issues, and relative wealth) were significant and were therefore included. The data satisfied all assumptions necessary for the logistic regression model to be valid.

Several relationships emerged within the final model, which considered low resilience as the reference (see Table 3). First, women in the middle and bottom PPS tertiles did not differ in their odds of being resilient (OR=0.99, p=0.91), while a comparison of those in the top and bottom tertiles yielded an OR of 2.12 (p<0.0001). Second, each single-year increase in age was associated with an OR for resilience of 1.014 (p<0.0001), while women with disabilities (OR=0.68, p=0.001) or mental health difficulties (OR=0.32, p<0.0001) were significantly less likely to be resilient. Third, resilience was negatively correlated with both longer lengths of displacement and poorer wealth relative to others in the community. Finally, only those who identified their ethnicity as 'other' exhibited significantly different, in this case decreased, odds of resilience compared to those who identified with none of the listed ethnicities, though the 1.30 OR associated with identifying as Afro-descendant was nearly significant (p=0.053).

## Discussion

This analysis elucidated a 65% prevalence of resilience, defined as those who felt they were able to cope 'All the Time'. However, drawing meaningful comparisons between this value and those reported in previous studies is challenging as resilience prevalence is rarely reported, particularly in studies about migrants. This issue is compounded by three pervasive patterns in the literature: (a) definitions of resilience are highly heterogeneous [28]; (b) measurement approaches are numerous, and inconsistently operationalize the construct; and (c) participants are seldom classified according to their level of resilience. For instance, it is common to define resilience as an absence of mental health disorders, such as depression and post–traumatic

**Table 2. Characteristics of high and low resilience migrants.**

| Variable | Category | Resilience | | | | P-Value |
| --- | --- | --- | --- | --- | --- | --- |
| | | All the Time (n=3872) | | Not All the Time (n=2118) | | |
| | | n | % | n | % | |
| **PPS** | | | | | | **<0.0001\*** |
| | Bottom Tertile | 1171 | 30.2 | 814 | 38.4 | |
| | Middle Tertile | 1189 | 30.7 | 819 | 38.7 | |
| | Top Tertile | 1512 | 39.0 | 485 | 22.9 | |
| **Ethnicity** | | | | | | 0.1886 |
| | Mestiza | 2137 | 58.3 | 1172 | 59.5 | |
| | Afro-Descendant | 236 | 6.4 | 106 | 5.4 | |
| | Indigenous | 180 | 4.9 | 92 | 4.7 | |
| | Other | 30 | 0.8 | 26 | 1.2 | |
| | None of These Apply | 1083 | 28.0 | 574 | 27.1 | |
| | Missing | 206 | 5.3 | 148 | 6.9 | |
| **Having a Child** | | | | | | **0.0090\*** |
| | Yes | 3241 | 84.1 | 1707 | 81.4 | |
| | No | 613 | 15.9 | 389 | 18.6 | |
| | Missing | 18 | 0.5 | 22 | 1.0 | |
| **Having a Partner** | | | | | | **0.0420\*** |
| | Married or in a Union | 1865 | 48.4 | 953 | 45.6 | |
| | Not Married or in a Union | 1990 | 51.6 | 1136 | 54.4 | |
| | Missing | 17 | 0.4 | 29 | 1.4 | |
| **Identifying as LGBTQ+** | | | | | | 0.9613 |
| | Yes | 104 | 2.8 | 57 | 2.8 | |
| | No | 3677 | 97.2 | 1999 | 97.2 | |
| | Missing | 91 | 2.4 | 62 | 2.9 | |
| **Length of Displacement** | | | | | | **<0.0001\*** |
| | < 1 Year | 1957 | 51.0 | 912 | 43.5 | |
| | 1 - 3 Years | 1119 | 29.2 | 664 | 31.7 | |
| | 3 - 5 Years | 669 | 17.4 | 460 | 22.0 | |
| | > 5 Years | 89 | 2.3 | 59 | 2.8 | |
| | Missing | 38 | 1.0 | 23 | 1.1 | |
| **Miscellaneous Health Issues** | | | | | | **<0.0001\*** |
| | Alcohol and Drugs | 20 | 0.5 | 14 | 0.7 | |
| | Disability | 246 | 6.4 | 166 | 7.8 | |
| | Mental Health Issues | 34 | 0.9 | 62 | 2.9 | |
| | None | 3572 | 92.3 | 1876 | 88.6 | |
| | Missing | 0 | 0.0 | 0 | 0.0 | |
| **Relative Wealth** | | | | | | **0.0002\*** |
| | Average or Above | 1359 | 36.7 | 648 | 31.3 | |
| | Poor | 1849 | 50.0 | 1132 | 54.8 | |
| | Very Poor | 492 | 13.3 | 287 | 13.9 | |
| | Missing | 172 | 4.4 | 51 | 1.0 | |

\* Indicates statistically significant difference between the high and low resilience groups for 2-sided Chi-square tests (α = 0.05)

**Table 3. OR estimates of resilience for the six variables in the final logistic regression model.**

| Variable | Category | OR | LCL | UCL | P-Value |
|---|---|---|---|---|---|
| **PPS** | Bottom Tertile (ref) | 1 | | | |
| | Middle Tertile | 0.99 | 0.87 | 1.14 | 0.91 |
| | Top Tertile | 2.12 | 1.83 | 2.45 | **<0.0001*** |
| **Age** | Per 1 Year Increase | 1.014 | 1.009 | 1.019 | **<0.0001*** |
| **Ethnicity** | None of These Apply (ref) | 1 | | | |
| | Afro-Descendant | 1.30 | 1.00 | 1.70 | 0.053 |
| | Indigenous | 1.06 | 0.80 | 1.42 | 0.68 |
| | Mestiza | 1.07 | 0.93 | 1.22 | 0.34 |
| | Other | 0.54 | 0.31 | 0.96 | **0.04*** |
| **Length of Displacement** | < 1 Year (ref) | 1 | | | |
| | 1 - 3 Years | 0.84 | 0.73 | 0.96 | **0.01*** |
| | 3 - 5 Years | 0.73 | 0.63 | 0.86 | **<0.0001*** |
| | > 5 Years | 0.76 | 0.53 | 1.09 | 0.13 |
| **Miscellaneous Health Issues** | None (ref) | 1 | | | |
| | Disability | 0.68 | 0.54 | 0.85 | **0.001*** |
| | Mental Health Issues | 0.32 | 0.21 | 0.50 | **<0.0001*** |
| | Alcohol and Drugs | 0.81 | 0.39 | 1.66 | 0.56 |
| **Relative Wealth** | Average or Above (ref) | 1 | | | |
| | Poor | 0.84 | 0.73 | 0.95 | **0.008*** |
| | Very Poor | 0.77 | 0.64 | 0.93 | **0.006*** |

\* Indicates statistically significant OR compared with the reference category (α = 0.05)

stress disorder. Using Blackmore et al.'s [3] systematic review and meta-analysis of studies undertaken across 15 countries as a guide, it could thus be argued that over two-thirds of refugees and asylum seekers are resilient. Alternatively, a study involving Syrian and Palestinian refugees in Lebanon found the participants' mean scores on the Connor-Davidson Resilience Scale 25 was 68.20±19.35 [79]. However, without a cut-off value above which participants were considered resilient, this figure cannot be meaningfully compared with the results of the present investigation. When contrasting our sample's 65% prevalence with the literature in general, the principal contributor to observed discrepancies is likely the characteristics of the current sample, which is unique as one which exclusively focused on Venezuelan migrant women. This likely limits the degree to which previous findings—often based on smaller, more specific, and contextually distinct samples of migrants—can be generalized. Social desirability bias may also have contributed to differential reports of resilience, as participants likely felt pressured to appear more in-control of their lives given the subject matter of the assessment. This interpretation is supported by the low percentage of participants who self-identified as LGBTQ+ (2.7%), which was notably below global and regional averages and likely reflects the significant stigma faced by sexually- and gender-diverse Venezuelan migrants [80].

When stratifying resilience by levels of PPS, women in the bottom and middle tertiles showed nearly identical resilience rates, suggesting the relationship between PPS and resilience to be more nuanced than originally anticipated. However, women in the top tertile were considerably more likely to report resilience, indicating a positive correlation which aligns with both this investigation's hypothesis and the literature base [28,30–40,42–49]. Mental health issues among participants were also unexpectedly infrequent, with a much smaller percentage reporting concerns than has been found in studies involving other migrant populations [5,43,81]. While the prevalence of mental health issues varies in the literature, even conservative estimates indicate higher rates among migrants, suggesting that social desirability bias, compounded by mental health stigma prevalent in Latin America [82], likely significantly contributed to underreporting.

Surprisingly, the bivariate analyses indicated neither LGBTQ+ self-identification nor ethnicity were significantly correlated with resilience. While the former may be partially explained by the small number of participants self-identifying, considering the degree of insignificance, additional factors are likely also at play, suggesting a complex relationship between LGBTQ+ self-identification and resilience that warrants further investigation. Unfortunately, the current lack of research on this dynamic among migrant populations restricts the degree to which this finding can be understood within the broader context of resilience research. Regarding ethnicity, in the final model only respondents who selected "other" exhibited significantly lower resilience; the positive association with identifying as Afro-descendant did not reach statistical significance. These findings suggest that the role of ethnicity in Venezuelan migrant women's resilience is complex and may be shaped more by cross-national sociopolitical contexts than by ethnicity alone. To illustrate, Peruvian authorities have been criticized for failing to ensure the safety of Venezuelan women by not guaranteeing their right to access justice and health services without discrimination [83], and the health system in general can be difficult for migrants to access [84]. In contrast, despite growing concerns about widespread xenophobia [85], Ecuador has adapted relatively well to address several areas of concern, including waving immigration fines for those who entered the country irregularly [86]. Alternatively, Brazil has fostered a highly supportive environment by facilitating voluntary relocation across the country [87] and allowing new arrivals to choose between migratory or refugee status, both of which confer the right to universal access to education and healthcare [88]. However, language barriers remain a challenge for Venezuela's predominantly Spanish-speaking migrants.

The LGBTQ +, child, and partner variables were each excluded from the final model as they did not exceed the $p < 0.20$ inclusion threshold. Collinearity between the latter two likely at least partially explains this finding; a conclusion which is supported by the strong correlation observed between the two factors, as removing the child variable greatly improved the statistical strength of the partner variable. In agreement with many [89,90], but certainly not all prior studies [32,42], resilience was positively correlated with age, although the effect size was small. Conversely, resilience was lower among women displaced for longer periods, although at >5 years the trend attenuated and was no longer significant. This suggests that, while eventual recovery is likely as migrants integrate more fully over time, this process may require several years to produce noticeable improvements following years of decline post-migration. Additionally, well-integrated migrants may have been underrepresented in the study, as they may have been less likely to be present at locations where sampling occurred, such as aid distribution centers, which could have biased the observed results. However, as we could not identify any applicable studies that examined this variable, additional research is needed to support or refute these theories.

Finally, the self-reported presence of disabilities and/or mental health issues, the use of alcohol and drugs, and poverty were significantly associated with lower resilience in the final model, highlighting the increased vulnerability of these groups. While evidence from recent systematic reviews corroborate the findings for mental health issues [3] and relative wealth [91], data on how resilience in migrant women relates to the use of alcohol and drugs and the presence of disabilities is minimal. However, existing evidence suggests that women and migrants living with disabilities are disproportionately exposed to a variety of potentially resilience-modulating factors. For instance, among women living in low- and middle-income countries, intimate partner violence is between two and four times more likely for those with disabilities compared to those without [92–94]. Furthermore, given the loss of support structures associated with migration, refugees with disabilities consistently face additional barriers to accessing supports and services, such as rehabilitation, assistive technologies, food, and healthcare [95], with those experiencing mental disabilities being particularly unlikely to be identified in registration and data collection efforts [96]. Individuals with disabilities are also more likely to experience violence generally, which may be due in part to physical and communication barriers, discrimination, and an increased need for regular assistance [97,98]. The consistent absence of information on potentially important factors in the resilience literature underscores the pressing need to expand the array of variables under examination in the field, especially through quantitative studies utilizing large sample sizes, which are scarce.

## Contributions and recommendations

This study provides valuable insights on the resilience of Venezuelan migrant women that could be used to inform policies in host nations. Given that including the five potential confounders in the model modified the PPS-resilience relationship by less than 2.5%, organizations could potentially identify migrants with low support and resilience through two simple questions: "How socially supported do you feel?" and "How often are you able to cope with the problems you face?" This approach would allow for the targeted allocation of resilience-boosting resources, in addition to time- and cost-effective screening processes, as early detection of high-risk individuals is particularly vital for fostering healthy adaptation and maximizing improvements to resiliency.

These findings can also be used to inform the development of resilience-promoting interventions for Venezuelan migrant women, and possibly for other populations of displaced South American women. Building on Chmitorz et al.'s [19] recommendation to assess resilience as an outcome rather than as a set of factors, potential interventions could focus on increasing migrants' perceived support and how frequently they feel they can successfully navigate adversity. For example, an intervention aimed at improving resilience among Venezuelan migrants in Brazil could target women with low resilience and low/mid PPS (since both were equally unlikely to be highly resilient) who are accessing migrant shelters or points of aid distribution. Using ecological momentary assessments, such as paper diaries, to capture daily stress levels with minimal recall bias, the program could begin with a brief period of self-assessment to establish a baseline. As guided by previous programs, the intervention could then be delivered through several group sessions, 60–90 minutes in duration, hosted in-person across multiple weeks. Previous interventions are highly heterogenous in their approaches, but most have involved training related to various combinations of mindfulness, psychoeducation, cognitive skills, self-compassion, gratitude, emotional regulation, relaxation, and/or goal setting [99]. Regardless of the approach selected, a program such as this should be designed from a strengths-based perspective that emphasizes migrants as strong and adaptable, while still acknowledging and accepting the abnormal frequency at which they experience potentially devastating adversity [100]. The support sessions could incorporate Venezuelan cultural elements, with the goals of reducing feelings of isolation, facilitating the sharing of coping strategies, and strengthening participants' social networks [101]. Participants should also be guided on how to effectively share their acquired resilience-building techniques with their communities to create a network of peer mentors [102], and post-program self-assessments could provide the opportunity for participants to reflect on their progress. At a systemic level, host nations should develop humanitarian settlement programs that are guided by positive psychology and adapted to the local context, including relevant factors such as language, culture, beliefs, and education. Combined, these guidelines would constitute a sustainable and cost-effective model to support resilience within Venezuelan migrant women facing ongoing hardship.

## Strengths and limitations

A principal strength of this study is its large sample size, which enhances confidence in its findings and allows for greater statistical power. Additionally, participants were sampled from nine locations across three countries, further improving the generalizability of the results by representing a diverse group of women. This study is also distinct in investigating the resilience of migrant women navigating a contemporary humanitarian crisis in South America—an area of study that has received limited attention [41,103]. Finally, employing sensemaking methodology in the design of the original survey mitigated the effects of two common biases; interpretation bias was minimized since participants interpreted their own experiences rather than relying on interviewer or researcher interpretations, and social desirability bias was reduced through the survey's choice of response options, which included all positive, negative, or neutral choices for any individual question. Together, these methodological approaches enhanced the validity of the data and the robustness of this study's findings.

A primary limitation of this analysis is that the dataset was not initially designed for resilience research, and thus information on certain, potentially important, confounders (e.g., religiosity) and contextual variables (e.g., immigration status) was not captured, thereby increasing the risk of uncontrolled confounding and restricting the degree to which the findings can be applied to

different contexts. For instance, legal immigration status was not collected in the original study because self-reported labels (e.g., "refugee") often diverge from statutory definitions, verification was infeasible, and the implementing partner (IOM) provides services irrespective of status. Consequently, interpretation of the PPS and displacement-duration variables is constrained by a lack of contextual information (e.g., whether migration was continuous and what protections/resources were legally available), limiting the utility of our results in supporting efforts to tailor recommendations to, for example, asylum seekers versus labor migrants. Another limitation is the use of convenience sampling, which introduced the potential for selection bias as women may have participated based on factors associated with both resilience and PPS, such as integration level and access to support networks. For instance, well-integrated migrants, who may be more likely to possess both robust resilience and highly proven support networks, could have been underrepresented if they were less likely to present in survey locations with high proportions of new migrants, such as points of aid distribution. Finally, as a cross-sectional study, this investigation is limited in its ability to describe the causal direction of the PPS-resilience relationship within this population.

## Conclusion

This study highlights the complex relationship between PPS and resilience among Venezuelan migrant women in Brazil, Ecuador, and Peru. Findings confirmed many expected relationships, such as a positive correlation between resilience and both PPS and relative wealth, as well as a negative association between resilience and the presence of disabilities, mental health issues, and/or the use of alcohol and drugs. Several unanticipated results were also uncovered, notably how the women in the lower two-thirds of PPS exhibited identical odds of resilience, and how LGBTQ+ self-identification was not significantly correlated with resilience, suggesting more nuanced relationships that warrant further research. Importantly, the odds of resilience in the model changed by less than 2.5% after adjusting for the five included potential confounders, indicating that Venezuelan migrant women can reliably report their levels of PPS and resilience with minimal need for additional contextualizing information.

This study also offers guidance for resilience-fostering interventions to support the mental well-being of Venezuelan migrant women. These insights can empower responders to more efficiently identify at-risk women and direct resources accordingly. It is also essential for future research endeavours to expand the existing array of resilience-modulating factors, particularly those providing contextual information relevant to the Venezuelan migration crisis. Understanding how these factors interact with resilience is essential for addressing the specific needs of Venezuelan migrant women and developing tailored supportive frameworks for their mental well-being throughout the years to come.

## Supporting information

**S1 Appendix. Survey questions and their coding.**
(DOCX)

**S2 Appendix. Parent survey questions.**
(DOCX)

**S1 Table. Regression summary.**
(XLSX)

**S1 Code. SAS code.**
(SAS)

## Acknowledgments

We would first like to thank all the participants who shared their migration experiences with the research team. We are grateful to all enumerators and country supervisors, as well as to the IOM team based in Panama City, including Monica Noriega. In Ecuador, the team included Ivone Vallejo, Francisco Toscano, Shirley Vélez, María Teresa Foyain, Melissa

Vindas, Mara Piedra, and Belén Rodríguez. The Peru team included Magda Carrera Abanto, Juan Cuba Del Río, Diana Lopez Orozco, Luis Alberto Carranza Calero, Maritza Villaseca Moran, Diana Reina Turpo Huichi, Aydee Yaquelin Mamani Olivera, Ari Jauregui, and Priscilla Silva. In Brazil, the team included Angel Jose Santana Valles, Albanelly López Rojas, Mayra Alejandra Figuera Ortiz, Solange Blanco, Orietta García, María M. García, Yelitza Lafont, William A. Clavijo Vitto, Blanca Montilla, Giulia Camporez, and Andressa Grechi. We thank all the supporting organizations that helped reach potentially marginalized participants in each location. We are also indebted to Laurie Webster of QED Insight for her extensive assistance with setup and data monitoring, in addition to our editor Katelyn Whytock for reviewing the manuscript. The funders of the original study possessed no role in study design, data collection/analysis, manuscript preparation, or the decision to publish.

## Author contributions

**Conceptualization:** Maxwell F. Josic, Bradley P. Stoner, Susan A. Bartels.

**Formal analysis:** Maxwell F. Josic.

**Funding acquisition:** Susan A. Bartels.

**Investigation:** Maria Marisol.

**Methodology:** Maxwell F. Josic, Bradley P. Stoner, Susan A. Bartels.

**Supervision:** Bradley P. Stoner, Susan A. Bartels.

**Visualization:** Maxwell F. Josic.

**Writing – original draft:** Maxwell F. Josic, Susan A. Bartels.

**Writing – review & editing:** Maxwell F. Josic, Bradley P. Stoner, Maria Marisol, Susan A. Bartels.

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
