## [Decision Letter · Decision Letter 0]

21 Mar 2025

PMEN-D-25-00043

The association between perceived psychosocial support and resilience among Venezuelan refugee and migrant women: A secondary analysis of cross-sectional data from 2022

PLOS Mental Health

Dear Dr. Bartels,

Thank you for submitting your manuscript to PLOS Mental Health. After careful consideration, we feel that it has merit but does not fully meet PLOS Mental Health’s publication criteria as it currently stands. Therefore, we invite you to submit a revised version of the manuscript that addresses the points raised during the review process.

Please note that we have only been able to secure a single reviewer to assess your manuscript. We are issuing a decision on your manuscript at this point to prevent further delays in the evaluation of your manuscript. Please be aware that the editor who handles your revised manuscript might find it necessary to invite additional reviewers to assess this work once the revised manuscript is submitted. However, we will aim to proceed on the basis of this single review if possible. 

Kindly attend to the expert reviewer's suggestions regarding greater clarity and refinement throughout, particularly concerning methodological details and sections of the introduction/discussion.

We look forward to receiving your revised manuscript.

Kind regards,

Avanti Dey, PhD

Staff Editor

PLOS Mental Health

Journal Requirements:

Additional Editor Comments (if provided):

Reviewers' comments:

Reviewer's Responses to Questions

**Comments to the Author**

1. Does this manuscript meet PLOS Mental Health’s publication criteria?

Reviewer #1: Partly

2. Has the statistical analysis been performed appropriately and rigorously?

Reviewer #1: Yes

3. Have the authors made all data underlying the findings in their manuscript fully available (please refer to the Data Availability Statement at the start of the manuscript PDF file)?

Reviewer #1: Yes

4. Is the manuscript presented in an intelligible fashion and written in standard English?

Reviewer #1: Yes

Reviewer #1: I would like to thank you for the opportunity to review this paper. It is a relatively under-studied topic within mental health of forcibly displaced communities, especially women. Thus, it is a pleasure to see such as big sample size and an innovative approach to investigate the contextualized resilience. Please see my comments for consideration below.

Abstract:

1. I’m not sure if the sentence “resilience protests against these threats..” is correct. The authors refer to migration experiences including sexual exploitation and trafficking as threats and resilience protects against these threats. Forcibly displaced people are resilient despite experiencing these threats/difficulties. I suggest that the authors revise this sentence.

2. Migrants and displacement are used together. It might be good to clarify whether the study includes Venezuelan migrants or Venezuelan forcibly displaced migrants (voluntary vs. involuntary nature of the migration).

3. What does “fully resilient” mean?

4. The authors stated that they are interested in the relationship between PSS and resilience, but the results do not include the result of that assumed relationship. They, then, concluded that this study showed PSS as important for resilience. I think they need to include the measures that they used for these two constructions in the study, PSS and resilience in the Abstract.

Introduction:

1. I suggest that the authors use forced displacement/forcibly displaced or refugees instead of migrants in the Manuscript.

2. Please include a reference for this sentence, Line 64: “Combined, these factors compound the risk of mental health issues developing..”

3. The first two paragraphs repeat each other. Especially, the Lines 57-63 with the Lines 67-80. These two paragraphs can be combined and the context can be condensed.

4. It is hard to follow the progression from greater risk of mental health due to forced displacement to resilience and state resilience as critical protective factor for mental health. From my perspective, despite all these challenges and increased risk for mental health and health related difficulties, most forcibly displaced communities do not develop mental health problems. In fact, they mostly are resilient. I would definitely pitch the story from the strength perspective, not the deficit. It also depends on how the authors define resilience, is it an outcome or a predictor?

There are resilience-promoting factors such as social support predicting resilience outcomes, which are better mental health/health outcomes than expected.

I strongly recommend the authors to read the following publications to help with their conceptualization:

Ungar, M., & Theron, L. (2020). Resilience and mental health: How multisystemic processes contribute to positive outcomes. The Lancet Psychiatry, 7(5), 441-448.

Schäfer, S. K., Supke, M., Kausmann, C., Schaubruch, L. M., Lieb, K., & Cohrdes, C. (2024). A systematic review of individual, social, and societal resilience factors in response to societal challenges and crises. Communications Psychology, 2(1), 92.

Ciaramella, M., Monacelli, N., & Cocimano, L. C. E. (2022). Promotion of resilience in migrants: a systematic review of study and psychosocial intervention. Journal of immigrant and minority health, 24(5), 1328-1344.

5. Reference needed for Line 95-96 & Line 110-113.

6. The authors mentioned that the Venezuelan refugee crisis is different from the other crisis settings. However, there is no contextual information about in what ways it is different.

7. Overall, I think the Introduction Section can benefit from clarification and refinement. The key issues are: 1) resilience and mental health are used together and interchangeably throughout the manuscript, the outcome variable is not clear. 2) the authors mentioned the importance of understanding the role of contextual factors on shaping resilience and social support, but it wasn’t stated as one of the objectives of the study.

After addressing these issues, I believe the Introduction will be strengthened by clearly defining the key concepts of the investigation and study objectives.

Methods:

1. The Sense-Making methodology is interesting and seems to address some pitfalls of the qualitative and quantitative methodologies. Has it been used in the forced displacement field before?

2. The respondents shared stories and answered the PSS question about themselves or other women/girls they might have known. On the other hand, coping ability, the outcome variable, was targeted at the respondent herself. How did the authors address the difference between the differential focus of these questions? What was their analytical approach for the answers for the stories not about the respondent herself?

3. Since the authors operationally defined resilience with the coping item, they need to provide a background in the Introduction that they would consider coping as the proxy measure of resilience.

4. Please include the name of the categories of PSS following categorizing the response options into tertiles, Line 200-201.

5. There needs to be a justification for categorizing the coping question into two. How did the authors consider “some of the time” or “never” as those with potential improvement?

6. Please include the name of the eight potential covariates, Line 211. Were they all categorical variables so that the authors used chi-square to examine the association between each of these variables and resilience (coping)?

7. Provide a reference for the choice of 0.20 as a threshold, Line 221.

Results:

1. The authors discussed the importance of contextualizing the experiences of Venezuelan women/girls; however, they haven’t used the relevant variables in the parent questionnaire, such as the star question, in their analyses. Which aspect of their analyses provided the contextualized experiences of the participants?

Discussion:

1. The reference 62 that the authors used to compare their results is with the Syrian refugee children, which does not provide a proper comparison. Two-thirds of the sample indicated probable resilience, and this estimate is comparable to those found in the extant literature with forcibly displaced populations. Most of the studies showed that approximately one third of the forcibly displaced individuals develop mental health issues (For instance: Blackmore, R., Boyle, J. A., Fazel, M., Ranasinha, S., Gray, K. M., Fitzgerald, G., ... & Gibson-Helm, M. (2020). The prevalence of mental illness in refugees and asylum seekers: A systematic review and meta-analysis. PLoS medicine, 17(9), e1003337.)

2. Line 374-377: The program components are not clear. Why workshops and why four weeks as the program duration? Please mention some of the evidence-based programs or strategies used with this population or a similar population to promote resilience and coping.

3. Line 395-396: Please provide references for “few studies”.

4. Overall, the authors provided the summary of their findings in the first two pages of their Discussion, however, the comparison of the resilience predicting factors with the extant literature is lacking in this section. How do the current findings complement/contradict the extant literature?

**Do you want your identity to be public for this peer review?** For information about this choice, including consent withdrawal, please see our Privacy Policy

Reviewer #1: No

---

## [Decision Letter · Decision Letter 1]

22 Jul 2025

PMEN-D-25-00043R1

The association between perceived psychosocial support and resilience among Venezuelan migrant women: A secondary analysis of cross-sectional data from 2022

PLOS Mental Health

Dear Dr. Bartels,

Thank you for submitting your revised manuscript to PLOS Mental Health. As communicated in my previous email, we needed to obtain a second review for your paper. Thank you for your understanding of this process and your patience. We now have a second report and would like to offer you one final round of revision. Please address the comments made by the reviewer, which you can see below. I will then assess the revisions in-house to ensure efficiency,

We look forward to receiving your revised manuscript.

Kind regards,

Karli Montague-Cardoso

Executive Editor

PLOS Mental Health

Journal Requirements:

Additional Editor Comments (if provided):

Reviewers' comments:

Reviewer's Responses to Questions

**Comments to the Author**

Reviewer #2: (No Response)

publication criteria?

Reviewer #2: Yes

3. Has the statistical analysis been performed appropriately and rigorously?

Reviewer #2: Yes

4. Have the authors made all data underlying the findings in their manuscript fully available (please refer to the Data Availability Statement at the start of the manuscript PDF file)?

Reviewer #2: Yes

5. Is the manuscript presented in an intelligible fashion and written in standard English?

Reviewer #2: Yes

Reviewer #2: Thank you for the opportunity to review this new version of the manuscript. Here's my detailed feedback and recommendations for improvement:

This manuscript presents research on the resilience of migrant women from Venezuela in some South American countries. While the topic is certainly relevant and the study aims to fill important gaps, there are still key areas that require significant clarification and expansion before it can be considered for publication.

The authors need to explicitly clarify the rationale for selecting participants aged "14+". This is crucial, as the legal implications of including minors (14-17 years old) must be thoroughly addressed, particularly concerning informed consent and their emancipation status within the specific legal frameworks of Brazil, Ecuador, and Peru. If legal emancipation was a criterion, it should be clearly stated and defined according to each country's laws or IOM definition.

One of the main variables is displacement duration, yet the participants are broadly described as migrants. The manuscript must differentiate between various legal and social categories of participants, such as refugees, asylum seekers, economic migrants, internally displaced persons, irregular migrants, or regular migrants. This distinction is vital because these statuses are directly linked to varying levels of vulnerability, access to resources, and consequently, mental health outcomes and resilience mechanisms. The current ambiguity significantly hinders the interpretation of findings related to "displacement duration." This was mentioned in the first review but was just reframed as migrant and this does not solve the problem.

The study involved interviews in different cities across Brazil, Ecuador, and Peru, but it largely omits a detailed discussion of the distinct socio-economic, political, and cultural realities within and between these countries. These contextual differences could significantly influence participants' experiences and resilience. While Lima is briefly mentioned, a more comprehensive analysis of these contextual factors is essential to enrich the findings and provide a nuanced understanding of the phenomenon under study. Such a comparative analysis would considerably strengthen the document.

The authors' reliance on their own previous publications for methodological discussions is a significant limitation. It is imperative to incorporate a broader range of external academic literature on the research methodologies, data analysis techniques (especially given the mention of different analytical approaches), and theoretical frameworks relevant to resilience studies or at least migration studies. Citing external, foundational texts would enhance the methodological rigor and scholarly credibility of the manuscript. Furthermore, while software company documentation might be useful for technical details, it should not serve as a primary methodological source and should be minimized or recontextualized within broader methodological discussions.

The statement on "68% of resilience in migrants" found on P22 L395, along with its associated citation, requires immediate clarification. The authors must explain the source, methodology, and context of this statistic. Is it a prevalence rate from a specific study, a general finding, or a measure derived from their own data? The current presentation is vague and potentially misleading.

Despite focusing on women participants, the manuscript does not consistently apply a clear gender perspective throughout the analysis and discussion. The authors should elaborate on how gender shapes the experiences of migration/displacement, resilience, and mental health for these women. Furthermore, the discussion regarding the exclusion of participants who do not identify as women, while simultaneously mentioning "LGBT+" as a variable, is confusing. The authors must clarify whether transwomen were included in the sample and, if so, how their experiences were considered within the "women" framework. They also need to explain the exact number and experiences of any LBT (Lesbian, Bisexual, Transgender) participants within the "women" sample, if applicable, and why "LGBT+" was initially considered as a variable if the study exclusively focused on women. The theoretical and analytical implications of these distinctions need to be thoroughly addressed.

The manuscript includes "disability" as a variable but does not define what this means within the context of the study or for the participating women. A deeper exploration of how disability intersects with migration/displacement and influences resilience for these women is warranted. This could include, for example, access to services, social support, and specific challenges faced.

The concluding statement asserting that "This study is also distinct in being the only of its kind to investigate the resilience of migrants navigating a contemporary humanitarian crisis in South America. Furthermore, based on our review of the literature, resilience studies that focus exclusively on migrant women are uncommon, and those that do often limit their samples to female subgroups, such as Sim’s work with Syrian mothers" is a significant overstatement. The authors must conduct a more thorough and comprehensive literature review, particularly considering studies published in Spanish and Portuguese, which are abundant in this field and region. The authors should revise this statement to accurately reflect the existing body of literature and articulate the specific unique contribution of their study within that broader context.

By addressing these points, the authors can significantly enhance the rigor, clarity, and scholarly contribution of their manuscript.

**Do you want your identity to be public for this peer review?** For information about this choice, including consent withdrawal, please see our Privacy Policy

Reviewer #2: No

---

## [Editor Report · Decision Letter 2]

18 Sep 2025

The association between perceived psychosocial support and resilience among Venezuelan migrant women: A secondary analysis of cross-sectional data from 2022

PMEN-D-25-00043R2

Dear Dr. Bartels,

We are pleased to inform you that your manuscript 'The association between perceived psychosocial support and resilience among Venezuelan migrant women: A secondary analysis of cross-sectional data from 2022' has been provisionally accepted for publication in PLOS Mental Health.

Best regards,

Karli Montague-Cardoso

Staff Editor

PLOS Mental Health